# *Ralstonia solanacearum* elicitor RipX Induces Defense Reaction by Suppressing the Mitochondrial *atpA* Gene in Host Plant

**DOI:** 10.3390/ijms21062000

**Published:** 2020-03-15

**Authors:** Tingyan Sun, Wei Wu, Haoxiang Wu, Wei Rou, Yinghui Zhou, Tao Zhuo, Xiaojing Fan, Xun Hu, Huasong Zou

**Affiliations:** 1State Key Laboratory of Ecological Pest Control for Fujian and Taiwan Crops, College of Plant Protection, Fujian Agriculture and Forestry University, Fuzhou 350002, China; sty15705905589@163.com (T.S.); weiwu120113@163.com (W.W.); 1170204039@m.fafu.edu.cn (H.W.); 18344936002@163.com (W.R.); zyh7426@163.com (Y.Z.); zhuotao@fafu.edu.cn (T.Z.); 000q351048@fafu.edu.cn (X.F.); xyq161352@163.com (X.H.); 2Fujian University Key Laboratory for Plant-Microbe Interaction, Fujian Agriculture and Forestry University, Fuzhou, Fujian 350002, China

**Keywords:** *atpA* gene, susceptibility, gene expression, RipX, *Ralstonia solanacearum*

## Abstract

RipX of *Ralstonia solanacearum* is translocated into host cells by a type III secretion system and acts as a harpin-like protein to induce a hypersensitive response in tobacco plants. The molecular events in association with RipX-induced signaling transduction have not been fully elucidated. This work reports that transient expression of RipX induced a yellowing phenotype in *Nicotiana benthamiana*, coupled with activation of the defense reaction. Using yeast two-hybrid and split-luciferase complementation assays, mitochondrial ATP synthase F1 subunit α (ATPA) was identified as an interaction partner of RipX from *N. benthamiana*. Although a certain proportion was found in mitochondria, the YFP-ATPA fusion was able to localize to the cell membrane, cytoplasm, and nucleus. RFP-RipX fusion was found from the cell membrane and cytoplasm. Moreover, ATPA interacted with RipX at both the cell membrane and cytoplasm in vivo. Silencing of the *atpA* gene had no effect on the appearance of yellowing phenotype induced by RipX. However, the silenced plants improved the resistance to *R. solanacearum.* Moreover, qRT-PCR and promoter GUS fusion experiments revealed that the transcript levels of *atpA* were evidently reduced in response to expression of RipX. These data demonstrated that RipX exerts a suppressive effect on the transcription of *atpA* gene, to induce defense reaction in *N. benthamiana*.

## 1. Introduction

The soil-borne pathogen *Ralstonia solanacearum* is a species complex that causes bacterial wilt disease in a wide range of host plants [1]. *R. solanacearum* isolates possess a meta-repertoire of type-III-dependent effectors, which comprise over 110 candidates that vary in different isolates [2,3]. In aids of transcription and translocation analysis, 72 effector candidates have been identified from the reference strain GMI1000 [4,5]. The effectors RipP1, RipAA1, and RipP2 specify the avirulent property to determine host range, which causes incompatible interactions with petunia, tobacco, or *Arabidopsis* [6,7,8]. RipAY, RipAL, RipAW, and RipAR show a suppressive effect on plant defense reactions [9,10,11]. On the other hand, transient expression of a few effectors induced hypersensitive response (HR)-like reactions in different plant species. For example, RipAX1 induces a hypersensitive response (HR) in eggplant, relying on a zinc protease motif [12].

The harpin-like protein RipX (previously named PopA) is purified from the culture supernatant of *R. solanacearum* GMI1000 and induces an HR in tobacco [13,14]. RipX can be processed into two additional active derivatives, RipX2 and RipX3, which lack 9 and 93 amino acids at the N-terminus, respectively [13]. In *R. solanacearum*, *ripX*, *ripAB*, and *ripAC* are organized in an operon that is under the control of HrpB [15]. Mutation in *ripX* retains full virulence in tomato plants and the ability to elicit an HR in tobacco plants [13,15]. The RipX protein is not only secreted outside by *R. solanacearum* but is delivered into host cells by a type III secretion apparatus [16]. Tobacco plants transformed with the *ripX* gene exhibited a promotion of resistance that depends on the salicylic acid (SA) signaling cascade. Transgenic plants inoculated with *Phytophthora parasitica* var. *nicotianae* show substantial upregulation of SA marker gene *PR1a* and the HR marker gene *hsr203J* [16]. In addition, the RipX protein shows Ca^2+^-dependent lipid binding and membrane integration characteristics, facilitating the formation of ion-conducting pores in the cell membrane [17].

Plant mitochondrial ATP synthase is a complex motor comprising a membrane-spanning F0 and a soluble F1 component [18]. F0 forms a proton turbine that is embedded in the inner membrane and connected to the rotor of F1. The F1 component protrudes into the mitochondria matrix, catalyzing the biosynthesis of ATP through the action of a rotational mechanism [19]. The assembly of the F1 component requires a module of the α3β3 hexamer and a central stalk γδε, aiding the nucleus-encoded but matrix-localized chaperones ATP11 and ATP12 [20,21]. Usually, subunit α (ATPA) has a molecular weight of 55 kDa and interacts with ATP12 to prevent unwanted homomeric α–α complexes [18]. The transcript abundance of the mitochondrial *atpA* gene is closely related to S male sterility in plants [22,23].

To gain an insight into the function of RipX, we combined yeast two-hybrid with split-luciferase and bimolecular fluorescence complementation assays, to characterize the interaction of RipX with *N. benthamiana* protein ATPA. Furthermore, the RipX showed a suppression effect on *atpA* gene transcription. This particular modulation mode provided a new clue to understand the signal transduction processes mediated by the elicitor RipX.

## 2. Results

### 2.1. Expression of RipX1 or RipX3 Induces a Defense Reaction in N. benthamiana

*Agrobacterium*-mediated transient expression was conducted to assess the response to RipX1 and RipX3 in *N. benthamiana*. The RipX3 protein was a truncated form of RipX1 obtained by deletion of the first 93 amino acids at the N-terminus (Figure 1A). At two days after agroinfiltration, the infiltrated leaves expressing either *ripX1* or *ripX3* showed yellowing phenotypes (Figure 1B). Simultaneously, cell death and hydrogen peroxide accumulation were found to be induced (Figure 1B). To ensure that the defense reaction was activated, the expression levels of the *hsr203J*, *hin1*, and *PR1a* genes were assayed. By comparison with plants transformed with empty vector, the three genes were found to be significantly induced in response to expression of either *ripX1* or *ripX3* (Figure 1C).

### 2.2. External Application of RipX1 and RipX3 Proteins Induces Macroscopic HR Reactions

To examine the HR-eliciting property, RipX1 and RipX3 were prokaryotically expressed in pET41a(+) fused with a GST tag. After purification, 0.2 μg/μL GST-RipX1 and GST-RipX3 proteins were infiltrated into *N. benthamiana* leaves. The results showed that both fusion proteins induced rapid macroscopic HR reactions 24 h post-infiltration (Figure 2A). Similar to the transient expression performance, the transcript levels of *hsr203J*, *hin1*, and *PR1a* were all elevated in *N. benthamiana* leaves that received GST-RipX1 and GST-RipX3 treatments (Figure 2B). To determine the minimal concentration essential for HR induction, the purified proteins were diluted to 0.1, 0.05, and 0.025 μg/μL. Infiltration of 0.1 and 0.05 μg/μL proteins induced obvious HR in *N. benthamiana* leaves. When a concentration of 0.025 μg/μL was applied, neither GST-RipX1 nor GST-RipX3 was able to induce macroscopic HR (Figure 2C). Thereafter, 0.05 μg/μL was the minimal concentration of RipX fusions used to induce macroscopic HR.

### 2.3. Isolation and Phylogenetic Analysis of RipX-Interacting Protein ATPA from N. benthamiana

In yeast two-hybrid (Y2H) assays, three positive clones were obtained from the *N. benthamiana* cDNA library, using RipX1 as bait. After the pGADT7 plasmids were extracted from yeast cells, one plasmid showed a positive interaction with RipX1 in the repeated Y2H assays. DNA sequencing revealed that the plasmid harbored a 2221-bp cDNA fragment that contains a 1527-bp open reading frame encoding the full-length mitochondrial *atpA* gene. The coding region was cloned into pGADT7 to repeat the interaction with RipX1 (Figure 3A). A split-luciferase assay was further performed to examine the interaction of RipX1 with ATPA (Figure 3B).

The full-length *atpA* gene encodes a 309-amino-acid ATPA protein (GenBank: MN365024). The amino acid sequence of this protein differed from that of the *N. tabacum* protein at only position 165, where arginine was changed to glutamine. The obtained *N. benthamiana* ATPA shared over 98% identity with the homologs from *Solanum pennellii*, *Solanum lycopersicum*, *Solanum pennellii*, and *Solanum tuberosum*. A phylogenetic tree established from 12 selected ATPA proteins revealed that the ATPA of *N. benthamiana* was branched into a monophyletic clade with homologs from *Solanaceae* plants (Figure 3C). Due to high conservation, the phylogenetic tree did not show distinctive diversity in *Solanaceae* plants. However, *Solanaceae* plant ATPAs showed distinctive differences from *Ipomoea nil*, *Humulus lupulus*, and *Arabidopsis thaliana* proteins (Figure 3C). In the genome database of *N. benthamiana*, there were 20 predicted cDNA transcripts showing similarity with the obtained ATPA (Appendix A). This finding indicated that there were multiple copies of ATPA in *N. benthamiana*.

### 2.4. RipX Interacts with ATPA at the Cell Membrane and Cytoplasm

RipX1 was fused to the C-terminus of red fluorescent protein (RFP) in the pGDR vector, and ATPA was fused to the C-terminus of yellow fluorescent protein (YFP) in the pGDY vector. RFP-RipX1 was localized to the cell membrane and cytoplasm, while YFP-ATPA was localized to the cell membrane, cytoplasm, and nucleus, as well as a few mitochondrial organelles (Figure 4A,B). RFP-RipX1 and YFP-ATPA were colocalized at the cell membrane and cytoplasm (Figure 4C). In an effort to detect the interaction in vivo, a bimolecular fluorescence complementation (BiFC) assay was employed to confirm that RipX1 and ATPA interacted at the cell membrane and cytoplasm (Figure 4D). In this case, the expressions of ATPA-YC and RipX1-YN were analyzed by Western blot detection (Figure 4E). To ensure the localization of ATPA to mitochondria, the mitochondrial marker MT-rk-CD3-991 was used. Upon transformation of the MT-rk-CD3-991 construct alone, a red fluorescence signal (RFP) was observed at the cell membrane, in the nucleus, and in a large number of mitochondria (Appendix A). When MT-rk-CD3-991 was co-transformed with GFP-ATPA, yellow fluorescence was observed in a few mitochondria, in addition to the cell membrane and nucleus (Appendix A). This finding indicated that ATPA did localize to mitochondria, although a large amount of ATPA was actually located in the cell membrane and nucleus.

### 2.5. Silencing of atpA Gene in N. benthamiana Has No Effect on HR Induction by RipX but Improves Resistance to R. solanacearum

Tobacco-rattle-virus-induced gene silencing (VIGS) was conducted to silence *atpA* in *N. benthamiana*. In comparison with the wild-type and *gfp*-silenced plants, the transcript level of *atpA* was reduced by 76% in *atpA*-silenced plants, indicating that the *atpA* gene was successfully disrupted (Figure 5A). In the *gfp*-silenced plants, transient expression of RipX1 or RipX3 induced a yellowing phenotype, and application of the RipX1 and RipX3 proteins induced macroscopic HR (Appendix A). This indicated that VIGS had no effect on defense induction by RipX. To know any deviation in host response to RipX, GST-RipX1 or GST-RipX3 was infiltrated in *atpA*-silenced plants. The results showed that external application of GST-RipX1 or GST-RipX3 caused macroscopic HR (Figure 5B). Simultaneously, silencing of the *atpA* gene showed no effect on RipX-induced yellow phenotype (Figure 5B). However, *R. solanacearum* caused a slowed disease development on *atpA*-silenced plants. In comparison with wild-type and *gfp*-silenced control plants, the disease index of *atpA*-silenced plants remained significantly low after inoculation. At 15 days post-inoculation, the disease index reached only 40%; meanwhile, wild-type and *gfp*-silenced plants were all dead (Figure 5C).

### 2.6. Expression of RipX1 Suppresses the Transcription of atpA Gene

The knock-down of *atpA* gene had no effect on the host response to RipX, and meanwhile promoted plant resistance to *R. solanacearum*. This prompted us to know the transcription pattern of *atpA* in response to RipX. Moreover, qRT-PCR analysis revealed that the transcript level of *atpA* was reduced by 52% in *N. benthamiana* expressing RipX1 relative to the uninoculated control (Figure 6A). In the *N. benthamiana* genome database, the contig Niben101Scf00438Ctg035 contains a 2357-bp DNA fragment upstream of the *atpA* gene, comprising a 789-bp partial sequence of the *cox3* (cytochrome oxidase subunit 3) gene. Four putative promoter elements were predicted from the *atpA* promoter region (Appendix A). To confirm the expression pattern of *atpA* affected by RipX, an *atpA* promoter GUS fusion was constructed in the pCAMBIA1381 vector. In histochemical staining analysis, promoter-driven GUS activity was reduced upon co-transformation with RipX1 (Figure 6B). The transcription of the promoter-driven *gusA* gene was additionally quantified by qRT-PCR. Expression of RipX1 led to a 72% reduction in *gusA* transcription compared with the expression of the promoter GUS fusion alone (Figure 6B).

## 3. Discussion

In the present study, we examined the defense reactions of *N. benthamiana* in response to purified RipX protein, as well as the transient expression of the *ripX* gene. Most importantly, the mitochondrial ATPA was identified as a RipX-interacting protein from *N. benthamiana.* Significant downregulation of the *atpA* gene was found in plants expressing *ripX*; meanwhile, the silencing of *atpA* gene improved the resistance to *R. solanacearum.* These results demonstrate that RipX induces defense reaction by suppressing the *atpA* gene, which plays a role in host susceptibility.

Harpins are heat-stable elicitors that induce HR when infiltrated into the leaves of tobacco and other plants [24]. Coupled with HR induction, a range of hormone signaling pathways are evoked to promote resistance to pathogens or insect attacks [25,26]. Notably, harpins are associated with altered mitochondrial functions [27]. HrpN treatment immediately leads to an inhibitory effect on ATP synthesis in tobacco cell cultures [27]. Heteroexpression of HrpZ*_Pss_* in the yeast *Saccharomyces cerevisiae* Y187 resulted in cell death but not in the “Petite” mutant, indicating the involvement of mitochondria in yeast cell death [28]. Here, we provide direct evidence that harpin-like RipX targets mitochondrial ATPA, to induce defense reactions in *N. benthamiana* plants.

Several harpins have been extensively characterized from plant-pathogenic bacteria since the first harpin HrpN was identified from *Erwinia amylovora* [24,29]. Harpins have been proven to form pores in liposomes of the cell membrane [24]. However, the harpins of diverse plant-pathogenic bacteria differ substantially in their primary structure, even though they carry relatively high amounts of glycine and serine residues [24]. This suggested that harpins are recognized by different host targets in the cell membrane. HrpN-interacting protein (HIPM) was cloned from *Malus* spp. and found to be localized to the plasma membrane [30]. *N. benthamiana* ATPA showed no similarity with HIPMs, which supported the conclusion that harpins target different host genes to execute their elicitor function. A recent study reported that HIPM acts as a susceptibility gene, as transgenic apple plants with disrupted HIPM expression showed a significant decrease in susceptibility to *E. amylovora* infection [31]. The *N. benthamiana atpA* gene is a potential susceptibility gene for bacterial wilt because the silencing of *atpA* led to increased resistance to *R. solanacearum*.

Because HR induction by harpin was initially discovered from external infiltration into plant tissue, it is widely accepted that harpins are perceived by the host plant as extracellular signals. However, emerging studies have demonstrated that harpin transgenic plants show enhanced resistance to pathogen attacks [16,32]. It appears that harpins can be perceived by the host plant inside of cells, although it is uncertain whether all harpins are translocated into plant cells during pathogen infection. By using a calmodulin-dependent adenylate cyclase reporter system, small amounts of the HrpN and RipX proteins have been successfully detected in plant-host cells [5,33]. In this study, transient expression of RipX induced a weak defense reaction in *N. benthamiana* cells. In contrast, external infiltration of the RipX protein induced macroscopic HR reactions. This discrepancy may be caused by the different perception mechanisms of *N. benthamiana* plants. In fact, we started to study the phenotype of transient expression of full-length RipX1 in *N benthamiana*. Because transient expression did not cause macroscopic cell death, a RipX3 construct was then used to confirm the phenotype. Thereafter, RipX induces diverse signaling pathways, either inside or outside of *N. benthamiana* cells.

In conclusion, our work shows that the *R. solanacearum* harpin RipX induces the defense reaction by modulating the mitochondrial *atpA* gene. When translocated into host cells through a type III secretion apparatus, RipX interacts with ATPA and exerts a suppressive effect on the transcription of the *atpA* gene. The particular mode provides details regarding the defense signaling pathways triggered by RipX.

## 4. Materials and Methods

### 4.1. Plant Material and Cultivation

*N. benthamiana* seeds were germinated in sterilized soil mix (peat moss:perlite (2:1, *v*/*v*)), in plastic pots. After germination, the seedlings were transferred to the Pindstrup Substrate (Pindstrup Mosebrug A/S, Denmark) potting mixture. Plants were grown in a greenhouse under a 16 h/8 h light/dark photoperiod, at 25 °C.

### 4.2. Bacterial Strains and Plasmids

All plasmids and strains used in this study are listed in Appendix A. *Escherichia coli* DH5α and *Agrobacterium tumefaciens* GV3101 were cultivated in Luria-Bertani medium, at 37 and 28 °C, respectively. *R. solanacearum* strain FJ1003 was grown at 28 °C in nutrient-rich broth medium [34]. Yeast strain AH109 was cultured in YPD medium (1% yeast extract, 2% peptone, and 2% glucose), at 30 °C. The 1035-bp *ripX1* and 750-bp *ripX3* were PCR amplified, to generate pHB:RipX1 and pHB:RipX3 for transient expression analysis. To study the subcellular localization, RipX1 was fused to RFP in pGDR, while RipX3 was fused to RFP in pGD-3G-mCherry. ATPA was fused to the C-termini of YFP and GFP in the vectors pGDY and pGDGm, respectively. To determine mitochondrial localization, the mitochondria targeting marker construct MT-rk-CD3-991 was co-transformed with GFP-ATPA. In the BiFC assays, RipX1 were cloned into 1301-YN for fusion to the N-terminal fragment of YFP. ATPA was fused with the C-terminal fragment of YFP in 1301-YC. The constructs were individually transformed into *A. tumefaciens* GV3101 for transient expression assays.

### 4.3. DNA Manipulation

DNA isolation, restriction enzyme digestion, and plasmid transformation were performed according to standard protocols [35]. The PCR primers used for molecular cloning, PCR product sizes, and detailed enzyme cleavage sites are listed in Appendix A.

### 4.4. Agrobacterium-Mediated Transient Expression

The cultured GV3101 cells were suspended in infiltration medium (10 mM MgCl_2_, 10 mM MES (pH 5.7), and 200 μM acetosyringone) and infiltrated into *N. benthamiana* leaves, using a needleless syringe. HR reaction, histochemical staining, and subcellular localization were examined 2 days after agroinfiltration.

### 4.5. Histochemical Staining

Cell death and hydrogen peroxide accumulation were examined by trypan blue and 3,3’-diaminobenzidine (DAB) staining, respectively. For trypan blue staining, *N. benthamiana* leaves were boiled in trypan blue solution for 20 min, kept at room temperature for 8 h, and then transferred into 2.5 g/mL chloral hydrate solution for de-staining by boiling for 25 min. For DAB staining, *N. benthamiana* leaves were immersed in 1 mg/mL DAB solution for 8 h and then cleared in 95% ethanol.

### 4.6. Confocal Microscopy

Leaf disks were harvested for fluorescence signal examination by a Leica confocal laser scanning microscope (SP8, Leica, Germany). After fluorescence signals were captured, the images were saved with a resolution of 1024 × 1024 pixels. For each construct or co-transformation analysis, three different samples were examined. Experiments were repeated three times.

### 4.7. Quantitative Real-Time PCR (qRT-PCR)

Total RNA was extracted from *N. benthamiana* leaves, using the Plant RNA Kit (Omega, Shanghai, China). RNA sample examination and cDNA synthesis were performed as described previously [36]. The primers are listed in Appendix A. The qRT-PCR experiments were performed with the following parameters: denaturation at 95 °C for 30 s, followed by 40 cycles of 95 °C for 5 s and 55 °C for 20 s. The expression of *EF1*α was used as an internal control. Relative expression levels and statistical analyses were conducted by CFX Maestro software [36]. Each experiment was repeated three times.

### 4.8. Examination of the HR Reaction Induced by RipX

The *ripX1* and *ripX3* genes were cloned into pET41a(+) for fusion with a GST tag (Appendix A). After the constructs were separately transformed into BL21(DE3), GST fusion proteins were induced with 0.1 mM isopropyl-β-d-thiogalactopyranoside and purified from the supernatant, using glutathione resin [36]. The purity and concentration of protein samples were analyzed by 10% SDS-PAGE and the Bradford method. Then, 0.2 μg/μL GST-RipX1 and GST-RipX3 proteins were infiltrated into *N. benthamiana* leaves, to detect HR induction. The GST tag was used as a negative control. To determine the minimum concentration essential for HR induction, the proteins were infiltrated into *N. benthamiana* leaves at 0.1, 0.05, and 0.025 μg/μL. At 24 h post-infiltration, detection of HR induction, histochemical staining, and qRT-PCR were conducted. Each performance was repeated three times.

### 4.9. Y2H

The construct pGBKT7:RipX1 and cDNA library of *N. benthamiana* were co-transformed into the yeast host strain AH109, by LiAc-mediated transformation. The pGADT7 plasmids were extracted from the screened positive colonies, to repeat the Y2H assay. The plasmid that exhibited a positive interaction with RipX1 was identified as ATPA based on DNA sequencing. The coding region of the *atpA* gene was amplified and cloned into pGADT7 to generate pGADT7:ATPA. After that, pGADT7:ATPA and pGBKT7:RipX1 were then co-transformed into AH109. The transformant was examined for growth on SD/-Ade/-Leu/-Trp/-His/ supplemented with 20 μg/mL X-α-galactosidase [36]. The Y2H repeat experiment was performed three times.

### 4.10. Split-Luciferase Complementation Assay

The coding sequence of RipX1 was cloned into pCAMBIA-CLuc to generate cLUC-RipX1. In addition, the coding sequence of ATPA was cloned into pCAMBIA-NLuc to generate ATPA-nLUC [37]. A mixture of *A. tumefaciens* GV3101 carrying the cLUC-RipX1 and ATPA-nLUC constructs was infiltrated into *N. benthamiana* leaves. The leaves were harvested 2 days post-agroinfiltration, rubbed with 0.5 mM luciferin, and kept in the dark for 5 min, to quench the fluorescence. A cooled charge-coupled device (CCD) imaging apparatus (Roper Scientific) was used to capture luciferase images.

### 4.11. Virus-Induced Gene Silencing (VIGS)

A 309-bp fragment of *atpA* was inserted into the pTRV2 vector. TRV2:*PDS* was used as the control, to evaluate the silencing efficiency [38]. TRV2:*gfp* carrying a 358-bp fragment of the *gfp* gene was used as a negative control. A mixture of GV3101 cultures (1:1 *v*/*v*) containing pTRV1 and pTRV2 constructs was co-infiltrated into 20-day-old *N. benthamiana* leaves. When newly grown leaves were fully expanded, qRT-PCR analysis was performed, to detect the transcripts of *atpA*. Next, transient expression of RipX1 and RipX3 was performed. Detection of the HR reaction, histochemical staining, and qRT-PCR analysis of defense genes were conducted 2 days post-agroinfiltration. Meanwhile, 0.05 μg/μL GST-RipX1 and GST-RipX3 were infiltrated into *atpA*-silenced plants. Detection of the HR reaction and qRT-PCR analysis of defense genes were conducted 24 h post-infiltration. The silencing experiments were repeated four times, with four independent plants for each repeat.

### 4.12. Resistance of N. benthamiana to Bacterial Wilt

The soil inoculation method was used to assess the disease development in *atpA*-silenced *N. benthamiana* plants [39]. The cultured cells of *R. solanacearum* were prepared to a final concentration of 1 × 10^8^ CFU/mL. The roots of wild-type, *atpA*-silenced or *gfp*-silenced control plants were first wounded and then were inoculated with *R. solanacearum* cell suspension, at 10 mL per pot. Disease severity and the percent severity index were recorded, using a previously described method [39]. This experiment was repeated four times. For every repeat, 6 wild-type, *atpA*-silenced, or *gfp*-silenced plants were assayed for disease development.

### 4.13. Promoter Activity Assays

The 1488-bp promoter region of *atpA* was cloned into pCAMBIA1381 in front of the *gusA* gene. The resultant pCAMBIA-P*atpA* construct was transformed into GV3101, for transient expression. GUS activity was studied by both histochemical staining and evaluation of *gusA* gene transcript levels 2 days after agroinfiltration. For histochemical staining, the infiltrated leaves were immersed in GUS staining solution (50 mM phosphate buffer (pH 7.0), 2 mM K_3_[Fe(CN)_6_], 2 mM K_4_[Fe(CN)_6_], and 0.1% 5-bromo-4-chloro-3-indolyl-β-d-glucuronide), at 37 ℃ overnight. Chloroplasts were removed with 70% ethanol, to produce a transparent background. The transcript level of the *gusA* gene was studied by qRT-PCR analysis. The experiments were repeated three times.

### 4.14. Western Blot Analysis

Total protein was extracted from *N. benthamiana* leaf disks by using Laemmli buffer at 2 days post-inoculation [40]. The proteins were then resolved by 12% SDS-PAGE and subjected to immunoblot analysis with anti-GFP and anti-Myc.

### 4.15. Sequence and Data Analysis

The amino acid sequence of ATPA was used to retrieve the sequences of 11 plant homologs deposited in GenBank. Twelve ATPAs from representative plant species were subjected to phylogenetic analyses, using the neighbor-joining (NJ) method implemented in MEGA5 [41]. The evolutionary distances were computed by using the Poisson correction method and are presented as the number of amino acid substitutions per site. The confidence probability (multiplied by 100) that the interior branch length is greater than 0, as estimated by using the bootstrap test (1000 replicates), is shown next to the branches. ATPA homologs in *N. benthamiana* were collected from the genome database (https://solgenomics.net/organism/Nicotiana_benthamiana/genome). The promoter sequence was predicted with http://www.fruitfly.org/seq_tools/promoter.html. Except for qRT-PCR analysis, the statistical analyses were conducted with SPSS 17.0 (SPSS Inc., Chicago, IL, USA). Statistically significant differences were conducted based on Tukey’s honest significant difference (HSD) test, using a one-way analysis of variance (ANOVA).

## Figures and Tables

**Figure 1 ijms-21-02000-f001:**
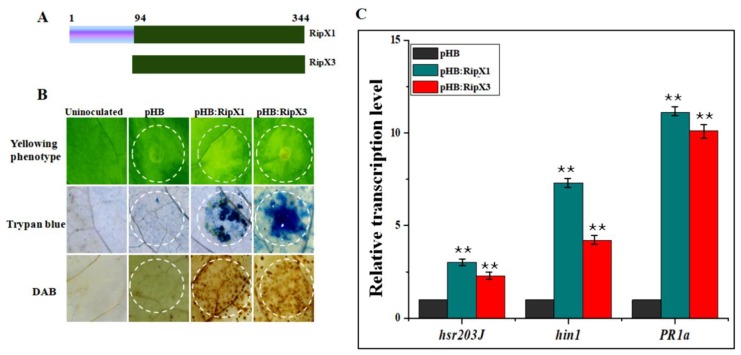
Defense reaction induced by transient expression of *ripX1* and *ripX3* in *Nicotiana benthamiana*. (**A**) Schematic diagram of expressed RipX1 and RipX3. (**B**) Yellowing phenotypes induced by RipX1 and RipX3. The infiltration areas were indicated by circles. All the experiments were repeated three times. (**C**) Activation of the defense genes *hsr203J*, *hin1*, and *PR1a*. The transcript level of each gene in plants transformed with empty vector pHB was set to 1, and levels in other plant samples were calculated relative to that. Data represent mean and standard deviation of three technical replicates. The asterisks denote statistical significance (** *p* < 0.01).

**Figure 2 ijms-21-02000-f002:**
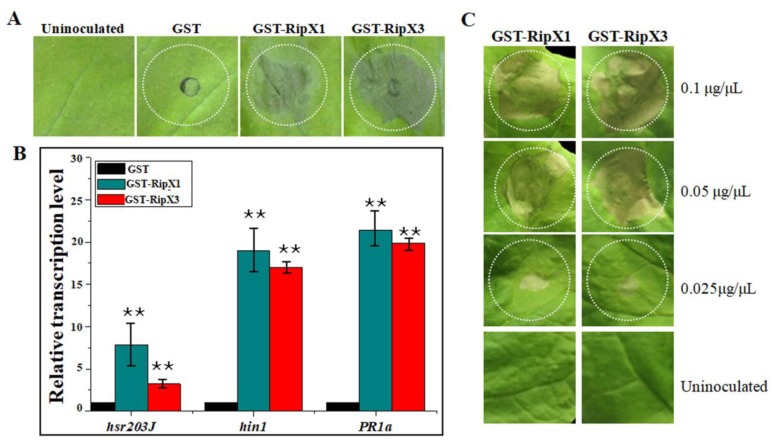
HR reactions induced by RipX1 and RipX3 proteins. (**A**) Macroscopic HR reaction induced by GST-RipX1 or GST-RipX3. A GST tag was used as a negative control. (**B**) Activation of the defense genes *hsr203J*, *hin1*, and *PR1a* examined by qRT-PCR analyses. The expression level for each gene in plants transformed with empty vector pHB was set to 1, and the level in other samples was calculated relative to that. Error bars indicate standard deviations of three individual replicates, and asterisks denote statistical significance (** *p* < 0.01). (**C**) Identification of the minimum concentration of the RipX protein essential for HR induction. All experiments were repeated three times.

**Figure 3 ijms-21-02000-f003:**
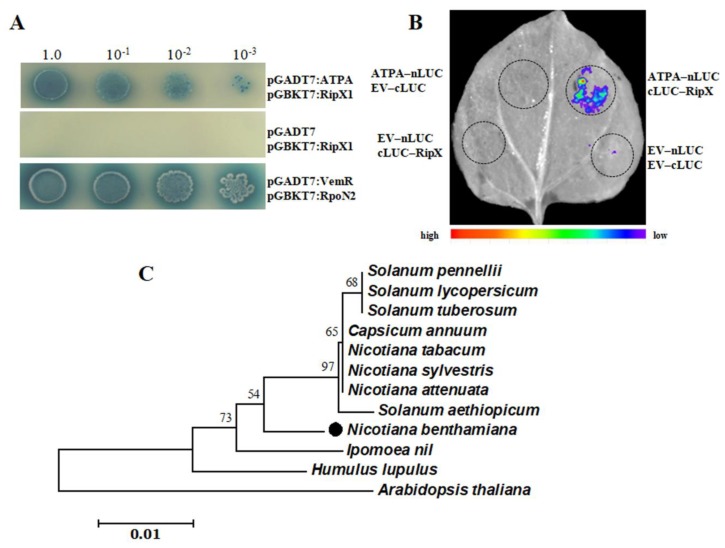
Identification of the ATPA protein that interacted with RipX1. (**A**) Y2H assays showing the interaction between ATPA and RipX1. The yeast strains were grown on SD/-Ade/-Leu/-Trp/-His/ plates supplied with X-α-galactosidase. The interaction between VemR and RpoN2 was used as a positive control. Y2H assays were repeated three times. (**B**) Split-luciferase assays of the interaction of ATPA with RipX1. This experiment was repeated three times, with similar results. Relative light units are shown below. (**C**) Evolutionary relationships of 12 ATPAs from representative plant species.

**Figure 4 ijms-21-02000-f004:**
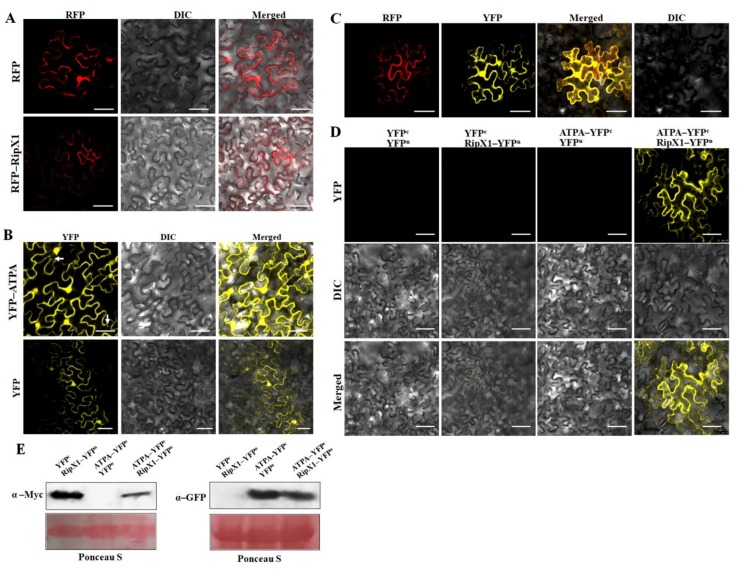
Subcellular localization of ATPA and its interaction with RipX1 in *N. benthamiana* cells. (**A**) Subcellular localization of RFP-RipX1. (**B**) Subcellular localization of YFP-ATPA. The arrows indicate the localization of YFP-ATPA in mitochondria. (**C**) Colocalization of RFP-RipX1 and YFP-ATPA in *N. benthamiana* cells. (**D**) BiFC analysis showing the spatial interaction of ATPA and RipX1 at the cell membrane. (**E**) Protein expression levels in the infiltrated plant leaves were collected for immunoblotting assays with anti-GFP and anti-Myc antibody. Ponceau S staining indicates the equal loading of the proteins. All experiments were repeated three times. The scale bar represents 50 μm.

**Figure 5 ijms-21-02000-f005:**
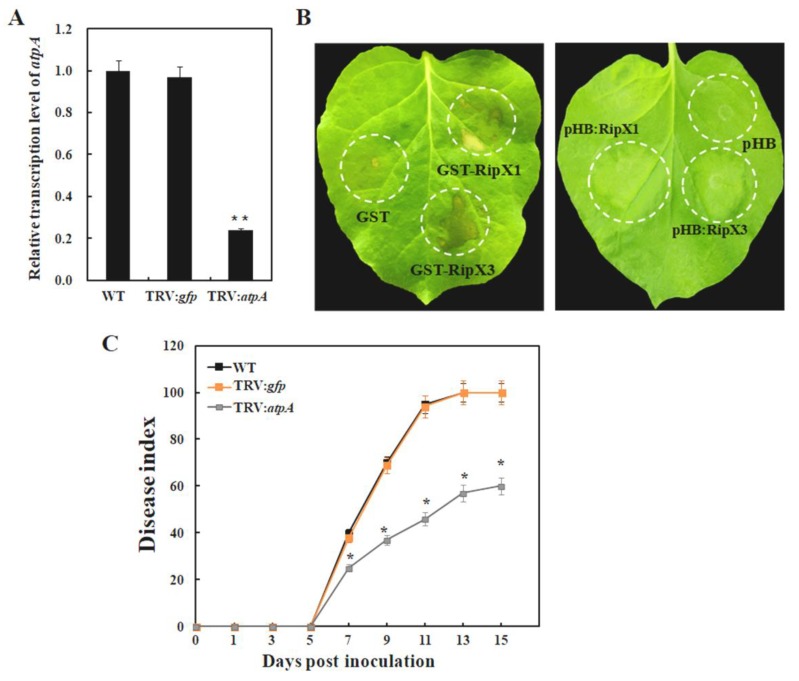
Involvement of the *atpA* gene in bacterial wilt disease. (**A**) Detection of the transcript level of *atpA* gene in gene-silenced plants by qRT-PCR. The transcript level in wild type was set to 1, and the level in other samples was calculated relative to that. Error bars indicate standard deviations of three individual replicates, and asterisks denote statistical significance (** *p* < 0.01). (**B**) Examination of the HR and yellowing phenotypes in *atpA*-silenced plants induced by RipX. (**C**) Disease development in *atpA*-silenced *N. benthamiana* plants. Each point represents the mean disease index of 24 plants combined from four separate experiments. Asterisks indicate the significant difference by ANOVA (*p* < 0.05).

**Figure 6 ijms-21-02000-f006:**
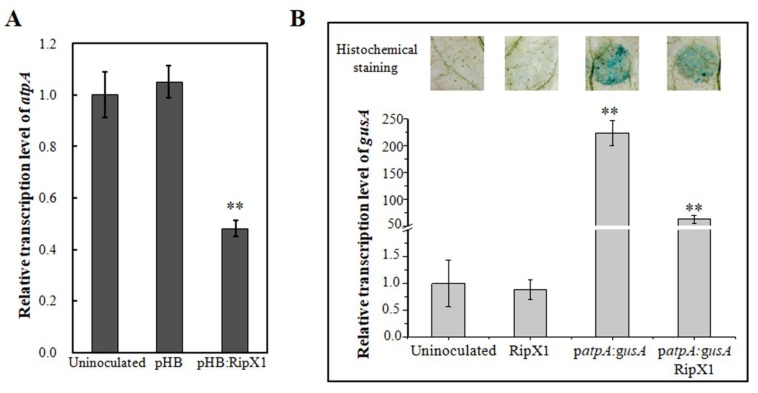
RipX1 suppressed *atpA* transcription. (**A**) Reduction in *atpA* transcription in *N. benthamiana* transiently expressing RipX1. The plants transformed with the pHB empty vector were used as controls. (**B**) Examination of *atpA* promoter activity. For qRT-PCR analyses, the transcript level in uninoculated plants was set to 1, and the level in other samples was calculated relative to that. Error bars indicate standard deviations of three individual replicates, and asterisks denote statistical significance (** *p* < 0.01).

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
