# Peer review of "Ralstonia solanacearum elicitor RipX Induces Defense Reaction by Suppressing the Mitochondrial atpA Gene in Host Plant"

_ijms, 2020, doi:10.3390/ijms21062000_

Round 1

Reviewer 1 Report

The authors carried out the current study “Ralstonia solanacearum elicitor RipX induces defense reaction by suppresssing the mitochondrial atpA gene in host plant “ identify the plant genes required by RipX to induce defense reactions and reported that RipX not only interacted with ATPA but also repressed atpA gene transcription. The study is on a topic of relevance and general interest to the readers of the journal. I found the paper to be overall well written and felt confident that the authors performed careful and thorough experiment and spectral processing. I have several significant concerns about the presentation of the data that should be addressed prior to publication.

 Abstract: Authors need to avoid using personal pronoun (eg. We, our ..etc

Material and Methods:

First 8 lines in 4.4. page 15 should be moved under section 4.2 in the same page

Section 4.7. should be quantitative Real-time PCR (qRT-PCR)

Section 4.8. line three is confusing, please make it clear about the method used

Section 4.10. complete the source for roper scientific, any abbreviation must associate with the full name at the first mention

Section 4.14. page 19 line 1: dpi, add the full definition

Results:

Page 4: Figure 1 caption contain extra information that will fit better if moved to either material and methods section or results section, for example I suggest Figure 1 caption to be “Figure 1. Defense reaction induced by transient expression of ripX1 and ripX3 in Nicotiana benthamiana. (A) Schematic diagram of expressed RipX1 and RipX3. (B) Yellowing phenotypes induced by RipX1 and RipX3. (C) Activation of the defense genes hsr203J, hin1 and PR1a. Data represent mean and standard deviation of three technical replicates. The asterisks denote statistical significance (**P Ë‚ 0.01).”

Page 5 and 6: Figure 2 caption, please follow the suggestion as in figure 1

Page 7 and 8: Figure 3 caption, please follow the suggestion as in figure 1

Follow the same suggestion for figure 4, 5, and 6

Author Response

Dear Reviewer, 

   Thank you for your work on our manuscript entitled “Ralstonia solanacearum elicitor RipX induces defense reaction by suppresssing the mitochondrial atpA gene in host plant” (ijms-731889). Your kind advices are valuable in improving the quality of our manuscript. We have studied comments carefully and have made corrections which we hope meet with approval. The revised words or sentences were marked with red color in addition to “Track Changes”. The main corrections in the paper and the responds to the reviewer’s comments are as followed:

Abstract: Authors need to avoid using personal pronoun (eg. We, our ..etc

Answer: “Here, we” was revised as “This work reported” (Page 1 Abstract section line 4)

“our” was revised as “These”. (Page 2 Abstract section line 16)

Material and Methods:

First 8 lines in 4.4. page 15 should be moved under section 4.2 in the same page

Answer: these sentences were moved to section 4.2. (Page 14 section 4.2 line 5-15)

Section 4.7. should be quantitative Real-time PCR (qRT-PCR)

Answer: “quantitative” was added. (Page 15 section 4.7 title)

Section 4.8. line three is confusing, please make it clear about the method used

Answer: The sentence in line 3 was revised as “After the constructs were separately transformed into BL21(DE3), GST fusion proteins were induced with 0.1 mM isopropyl-β-D-thiogalactopyranoside and purified from the supernatant using glutathione resin [36]. The purity and concentration of protein samples were analyzed by 10% SDS-PAGE and the Bradford method.” (Page 15 section 4.8 line 2-6)

Section 4.10. complete the source for roper scientific, any abbreviation must associate with the full name at the first mention

Answer: full name is given as “charge-coupled device (CCD)….” (Page 16 Section 4.10 line 7)

Section 4.14. page 19 line 1: dpi, add the full definition

Answer: “dpi” was revised as “days post inoculation”. (Page 18 Section 4.14 line 2)

Results:

Page 4: Figure 1 caption contain extra information that will fit better if moved to either material and methods section or results section, for example I suggest Figure 1 caption to be “Figure 1. Defense reaction induced by transient expression of ripX1 and ripX3 in Nicotiana benthamiana. (A) Schematic diagram of expressed RipX1 and RipX3. (B) Yellowing phenotypes induced by RipX1 and RipX3. (C) Activation of the defense genes hsr203J, hin1 and PR1a. Data represent mean and standard deviation of three technical replicates. The asterisks denote statistical significance (**P Ë‚ 0.01).”

Answer: Figure 1 caption was revised as suggested. (Page 4)

Page 5 and 6: Figure 2 caption, please follow the suggestion as in figure 1

Answer: Revised. Deleted “The prokaryotically expressed GST-RipX1 and GST-RipX3 proteins were diluted to 0.2 μg/μl and then infiltrated into N. benthamiana leaves.”; “The transcript levels were evaluated 24 hours after infiltration.” (Page5-6)

Page 7 and 8: Figure 3 caption, please follow the suggestion as in figure 1

Answer: Deleted the sentences describing material or methods (Page7).

The phylogenetic analysis in MEGA5 was moved to in Material and Methods. (Page 18 Section 4.15 line 4-8)

Follow the same suggestion for figure 4, 5, and 6

Answer: Deleted the sentences describing material or methods (Page 8-9, 10,11)

Reviewer 2 Report

The research described in this manuscript is very interesting and provides new findings about the defense response of Nicotiana benthamiana against the pathogen R. solanacearum.

However, in my opinion the manuscript requires minor revisions.

It's a hard one to read. I suggest editing the text to get a better understanding for the reader.

Ralstonia (formerly Pseudomonas) solanacearum is a major bacterial phytopathogen, causing lethal wilting in more than 200 plant species worldwide (Denny, 2000). This pathogen secretes more than 70 type III effector proteins called Rips (Ralstonia-injected proteins) into plant cells at an early stage of infection to succeed in infection. Recent studies revealed several R. solanacearum effectors that can suppress plant defense responses in various ways. However, the effect of other effectors on plant immunity remains unclear such as RipX (formerly PopA). This effector is released by type III secretion from the bacterial plant pathogen Ralstonia solanacearum and triggers the hypersensitive response (HR) in tobacco. The function of PopA remains obscure, mainly because mutants lacking this protein are not altered in their ability to interact with plants.

In this manuscript, the authors showed elicitor RipX of R. solanacearum exerts a suppressive effect on the transcription of atpA gene to induce defense reaction in Nicotiana benthamiana. My question is, will the same thing happen with other types of plants like tobacco?

The manuscript is a basically well-organized paper with interesting data and encourage you to use the comments to improve your manuscript.

Key words Authors should select new key words which are not included in the title. All the chosen ones are in the title.

In the introduction I miss talking a bit more about the hypersensitive response in plants. In addition, I think it is important to bring out the importance of the genes whose expression is analyzed in this manuscript.

I find the overall objective of this work “The purpose of this research was to identify the plant genes required by RipX to induce defense reactions”, to be too ambitious, I would suggest not doing it so broadly but more precisely.

Results

I suggest modifying the headings of each point in the results because more than a heading is the result of that point. I find it more logical to put the headings that you can read in the figures and which match more to materials and methods.

Could you clarify the reasons for selecting RipX1 for the tests described in point 2.3.1?

In the figures you should show indicate statistical analysis and number of replicates.

In figure 2S the names PopA1 and PopA3 should be changed to RipX1 and RipX2, respectively.

Discussion

 I suggest to begin the discussion with the most conclusive results found and I would close it with the conclusion/s more remarkable of this work. This is exactly where it will lead and what it is useful for.

On the other hand, the discussion contains sections that are redundant with the results and should be omitted.

Author Response

Dear Reviewer,   

   Thank you for your work on our manuscript entitled “Ralstonia solanacearum elicitor RipX induces defense reaction by suppresssing the mitochondrial atpA gene in host plant” (ijms-731889). Your kind advices are valuable in improving the quality of our manuscript. We have studied comments carefully and have made corrections which we hope meet with approval. The revised words or sentences were red colored, except for “Track Changes”. The main corrections in the paper and the responds to the reviewer’s comments are as followed:

In this manuscript, the authors showed elicitor RipX of R. solanacearum exerts a suppressive effect on the transcription of atpA gene to induce defense reaction in Nicotiana benthamiana. My question is, will the same thing happen with other types of plants like tobacco?

Answer: Thank you for your suggestion. We did want to know whether the atpA gene is involved in susceptibility in tomato and eggplant. Based on the analysis of sequence similarity, a VIGS construct will be used to study the disease development on gene silencing plants.

Key words Authors should select new key words which are not included in the title. All the chosen ones are in the title.

Answer: The key words were revised as “atpA gene, susceptibility, gene expression, RipX, Ralstonia solanacearum”. (Page 2 line 8-9 )

In the introduction I miss talking a bit more about the hypersensitive response in plants. In addition, I think it is important to bring out the importance of the genes whose expression is analyzed in this manuscript.

Answer: The defense genes of PR1a and hsr203J were added in the introduction to show the importance related with RipX: “Transgenic plants inoculated with Phytophthora parasitica var. nicotianae show substantial upregulation of SA marker gene PR1a and the HR marker gene hsr203J”  (Page 3 line 4-6)

I find the overall objective of this work “The purpose of this research was to identify the plant genes required by RipX to induce defense reactions”, to be too ambitious, I would suggest not doing it so broadly but more precisely.

Answer: The sentences was revised as “To gain an insight into function of RipX, we combined yeast two-hybrid with split-luciferase and bimolecular fluorescence complementation assays to characterize the interaction of RipX with N. benthamiana protein ATPA. Furthermore, the RipX showed a suppression effect on atpA gene transcription.” (Page 3 line 20-23)

Results

I suggest modifying the headings of each point in the results because more than a heading is the result of that point. I find it more logical to put the headings that you can read in the figures and which match more to materials and methods.

Answer:  heading 2.3 was revised as “2.3. Isolation and Phylogenetic Analysis of RipX-interacting Protein ATPA from N. benthamiana” (Page 6 line 5-6)

Headling 2.5 was revised as “Silencing of atpA Gene in N. benthamiana Has No Effect on HR Induction by RipX but Improves Resistance to R. solanacearum” (Page 9 line 8-9)

Could you clarify the reasons for selecting RipX1 for the tests described in point 2.3.1?

Answer: We had no particular reason to use RipX1 to screen interacting protein, just use a full length.  

In the figures you should show indicate statistical analysis and number of replicates.

Answer: For all the qRT-PCR analysis, statistical analysis was performed by CFX Maestro software (Biorad). Statistical analysis was added in disease index data in Figure 5 caption: Asterisks indicate the significant difference by ANOVA (P < 0.05). (Page 10)

In figure 2S the names PopA1 and PopA3 should be changed to RipX1 and RipX2, respectively.

Answer:  Revised. Please see in our resubmitted data.

Discussion

 I suggest to begin the discussion with the most conclusive results found and I would close it with the conclusion/s more remarkable of this work. This is exactly where it will lead and what it is useful for.

 Answer: One paragraph was added into discussion to give the most conclusive results. Please determine whether we heat the points. (Page 12 line 2-8)

On the other hand, the discussion contains sections that are redundant with the results and should be omitted.

Answer: The third paragraph in original manuscript was deleted from discussion section, including the reference 29 and 30. In accordingly, the reference citation in the manuscript was revised in order. (Page 13 line 4)